# Morphological and Molecular Identification of *Sarcocystis arctica* in Captive Cheetahs (*Acinonyx jubatus*) in China Helps Clarify Phylogenetic Relationships with *Sarcocystis caninum* and *Sarcocystis felis*

**DOI:** 10.3390/ani15020180

**Published:** 2025-01-10

**Authors:** Zhe Liao, Niuping Zhu, Yurong Yang, Shuangsheng Deng, Thomas Jäkel, Junjie Hu

**Affiliations:** 1Yunnan Key Laboratory for Plateau Mountain Ecology, Restoration of Degraded Environments, School of Ecology and Environmental Sciences, Yunnan University, Kunming 650091, China; liaozhe@mail.ynu.edu.cn; 2College of Veterinary Medicine, Henan Agricultural University, Zhengzhou 450046, China; zhuniuping414@163.com (N.Z.); yangyu7712@sina.com (Y.Y.); 3Joint Laboratory of Virology & Immunity, School of Biological Sciences, Yunnan University, Kunming 650091, China; ssdeng@ynu.edu.cn; 4Institute of Biology, Department of Parasitology, University of Hohenheim, 70599 Stuttgart, Germany

**Keywords:** carnivore intermediate host, ultrastructure, nuclear 18S rRNA, 28S rRNA, ITS-1, mitochondrial *cox*1, phylogenetic analysis

## Abstract

*Sarcocystis* spp. are cyst-forming, intracellular protozoan parasites that are transmitted between hosts through a predator–prey relationship. Cheetahs (*Acinonyx jubatus*) are unique among felids due to their distinct craniodental and skeletal morphology. There was only one previous report describing cysts of *Sarcocystis* in the musculature of cheetahs; these sarcocysts were designated *S. felis* because of morphological similarity. Meanwhile, however, it was observed that sarcocysts of *S. felis* were morphologically indistinguishable from those of *S. arctica* and *S. caninum* in canids. *Sarcocystis arctica* and *S. caninum* have been regarded as potentially conspecific based on high similarity between genetic markers. Here, *Sarcocystis* sarcocysts were observed in two captive cheetahs after they had died in zoos in Zhengzhou city, China. Based on morphological observations and analyses of four marker genes, we identified our parasite samples as *S. arctica*, being closest to this species that was originally observed in wild canids, rather than *S. felis*, which was genetically more distant. The occurrence of *S. artica* of canids as intermediate hosts diagnosed in a feline species suggests a broader intermediate host range for this *Sarcocystis* species for which predatory birds are likely definitive hosts. Furthermore, our study supports the notion of conspecifity of *S. caninum* with *S. arctica*.

## 1. Introduction

Species of the genus *Sarcocystis* are tissue cyst-forming intracellular protozoan parasites with an obligate, heteroxenous life cycle that is based on a predator–prey relationship between a definitive host (predator) and an intermediate host (prey). In the intermediate host, asexual reproduction generates sarcocysts in the striated musculature; in the definitive host, intestinal oocysts/sporocysts are produced through sexual reproduction and shed in feces [1].

The cheetah (*Acinonyx jubatus*) is unique among felids because of its distinct craniodental and skeletal morphology; it inhabits African grasslands and semideserts [2]. Previously, a single report described tissue cysts of *Sarcocystis* in the musculature of captive cheetahs in North America, and these sarcocysts were designated as *S. felis* because of their morphological similarity to this species [3]. *Sarcocystis felis* was initially named for its sarcocysts detected in bobcats (*Felis rufus*, synonym *Lynx rufus*), domestic cats (*F. catus*), and cougars and panthers (*F. concolor*) [4]. Lions (*Panthera leo*) were subsequently added to the list of intermediate hosts of this parasite [5]. However, all these identifications depend mainly on morphological similarities between the respective sarcocysts. Interestingly, sarcocysts morphologically indistinguishable from those of *S. felis* were also found in domestic and wild canids and named *S. arctica* in the arctic fox (*Vulpes lagopus*) [6,7] and *S. caninum* in the domestic dog (*Canis lupus familiaris*) [8]. *Sarcocystis caninum* and *S. artica* appear to be identical at the molecular level [9,10], but to date, both species have been treated as separate species [1,11].

In view of these uncertainties surrounding the identity of *Sarcocystis* species in canid and felid intermediate hosts, molecular methods have become increasingly important for the identification and delimitation of species [12] and, ultimately, for assisting in the effective control and prevention of *Sarcocystis* infections in wild and domestic animals. At present, there are various nucleotide sequences of *S. felis* from domestic and wild felids, and *S. arctica* and *S. caninum* from domestic and wild canids, available in GenBank. However, owing to the relatively small genetic distances between the various taxa within what we call the *S. neurona* cluster of species [13], including the species under investigation, it is often difficult to fully resolve the branching patterns and, hence, phylogenetic relationships among these taxa. For example, this can result in limited tree resolution (e.g., polytomy) and low bootstrap support for markers such as the nuclear 18S rRNA gene but also impact faster-evolving sequences such as ITS-1 if fragments are small, e.g., [9,14,15]. Long-branch attraction (LBA) caused by large differences in evolutionary rates between different lineages of *Sarcocystis* may even distort tree topology in that Toxoplasmatinae become nested within Sarcocystinae, e.g., [6].

Here, we diagnosed for the first time sarcocysts in the musculature of cheetahs that were held under human care and had died in zoos in China. Based on the morphological characterization of these sarcocysts and genetic analysis of four marker genes (nuclear 18S rRNA, 28S rRNA, ITS-1, and mitochondrial *cox*1), we aimed to identify the *Sarcocystis* species at hand. We reassessed the phylogenetic relationships among the most closely related taxa, i.e., *S. arctica*, *S. caninum*, and *S. felis*, via secondary structure-guided alignment of all ribosomal sequences to improve tree resolution and focused on a cluster of *Sarcocystis* spp. that mainly use birds as hosts. Recently, raptors or scavenging birds were shown to serve as definitive hosts for the development of *Sarcocystis* spp. in the musculature of mammalian carnivores [16,17]. Finally, we discuss the potential epidemiological and ecological factors that might explain natural infection in cheetahs and carnivore predators in general.

## 2. Materials and Methods

### 2.1. Tissue Sampling and Morphological Examination of Sarcocysts from Cheetahs

Samples were taken from one female (#1) and five male (#2–#5) cheetahs, who had died at two zoos in Zhengzhou city, located in central China, from 2017 to 2022. All the animals had been imported from Africa and maintained in captivity for 4 to 21 years. The carcasses were shipped to the veterinary pathology laboratory of Henan Agricultural University for pathological examination. The pathological findings were published by others: Briefly, histopathological examination revealed chronic renal insufficiency (4/5), interstitial pneumonia (2/5), necrotic splenitis (1/5), splenatrophy (2/5), septic spleen (1/5), systemic atrophy (1/5), arterial sclerosis (1/5), and acute endometritis (female cheetah), whereby 3 of the 5 males were diagnosed to have died of renal insufficiency and the other 2 by natural aging. Additionally, serological examination indicated that all six animals were infected with *Toxoplasma gondii* and viable *Toxoplasma* parasites could be retrieved from muscle tissue through bioassays in mice. However, none of the cheetahs showed any acute parasite infection [18]. In the laboratory, fresh muscular tissues collected from each animal (leg muscle, tongue, diaphragm, esophagus, and heart) were squeezed between two glass slides and inspected for sarcocysts via a stereomicroscope. Sarcocysts were isolated from muscular fibers and directly observed via light microscopy (LM) or processed for a DNA extraction and analysis. Small muscular pieces (ca. 1 cm × 5 mm × 5 mm; length × width × thickness, respectively) containing sarcocysts were placed separately into 10% buffered formalin and 2.5% buffered glutaraldehyde for fixation and subsequent sectioning and transmission electron microscopy (TEM). Formalin-fixed muscle was processed via routine methods and embedded in paraffin wax. Sections (4 mm) were cut and stained with hematoxylin and eosin (H&E). Small portions of glutaraldehyde-fixed muscle were processed for TEM as described previously [19]. Ultrathin sections were stained with uranyl acetate and lead citrate and then examined via a JEM100-CX transmission electron microscope. For DNA extraction, individual sarcocysts were stored in sterile water at −20 °C prior to processing.

### 2.2. DNA Isolation, PCR Amplification, Cloning, and Sequence Analysis

Four individual sarcocysts, two isolated from Cheetah #5 and two isolated from Cheetah #6, were separately subjected to genomic DNA extraction via a TIANamp Genomic DNA Kit (Tiangen Biotech Ltd., Beijing, China), according to the manufacturer’s instructions. The *Sarcocystis* species were characterized at four loci located in 18S rRNA, 28S rRNA, ITS-1, and the mitochondrial *cox*1. 18S rRNA was amplified with the primer sets ERIB1/S2 and S3/B [20,21,22]; 28S rRNA was amplified with the primer sets KL1/KL3, KL4/KL5b, and KL6/KL2 [23]; ITS-1 was amplified with the primer pairs JNB69/JNB70 [24]; and mitochondrial *cox*1 was amplified with the primer pairs SF1/SR9 [12,25]. Polymerase chain reaction (PCR) amplification was performed in a 25 μL reaction with 1× PCR buffer, 0.15 mmol MgCl_2_, 0.25 mmol dNTPs, 1 U Taq DNA polymerase (TakaRa, Dalian, China), 50–100 ng of DNA, and 25 pmol of each primer. PCR amplification was performed as previously described [25]. The resulting PCR products were gel-purified via an E.Z.N.A.^®^ Gel Extraction Kit (Omega BioTek, Inc., Norcross, GA, USA) and ligated to the pCE2 TA/Blunt-Zero vector using a 5 min TA/Blunt-Zero Cloning Kit (Vazyme Biotech Co., Ltd., Nanjing, China) according to the manufacturer’s instructions. The ligated vectors were subsequently transformed into Trelief^®^ 5α Chemically Competent Cells (Tsingke Biotechnology Co., Ltd., Beijing, China). The positive bacterial clones were sequenced in both directions by an ABI PRISM TM 3730 XL DNA Analyzer (Applied Biosystems, Thermo Fisher Scientific, Waltham, MA, USA).

The sequences were assembled via multiple overlapping regions using the SeqMan II program (DNAStar, Madison, WI, USA). The pairwise identity/similarity among the newly obtained sequences was calculated via the program BioEdit [26]. The initial GenBank screening and pairwise comparison of the newly obtained sequences were performed using the web-based Basic Local Alignment Search Tool (BLASTn) of the National Center for Biotechnology Information of the National Institutes of Health, Bethesda, MD, USA.

### 2.3. Phylogenetic Analysis

Phylogenetic analyses of nucleotide sequences were conducted separately for nuclear 18S rRNA, 28S rRNA, and ITS-1 and mitochondrial *cox*1 using MEGA 11 software [27]. The sequences of the four genes of related *Sarcocystis* spp. were downloaded from GenBank, whereby in our analyses, we focused on the closest *Sarcocystis* relatives of the samples under investigation, namely, *S. arctica*, *S. caninum*, *S. felis*, and *S. canis*, for which we included different haplotypes (if available) to allow for variability and check their potential influence on tree topology. With respect to other *Sarcocystis* spp., we included mainly those with birds as definitive or intermediate hosts plus species closely related to *S. neurona*. *Sarcocystis* spp. of ruminants as intermediate hosts were excluded because these taxa form a well-established, separate lineage with significantly higher evolutionary rates than the *Sarcocystis* species under investigation. This could interfere with tree topology, leading to undesirable effects such as long-branch attraction [13]. Members of the subfamily Toxoplasmatinae served as outgroups. The ribosomal RNA and spacer sequences were aligned via the multiple sequence alignment algorithm of the “R-Coffee” web server, namely, a combination of the methods MAFFT, MUSCLE, and ProbconsRNA, which considers the predicted secondary structure of RNA [28]. The alignments were subsequently checked visually and truncated at both ends so that most sequences started and ended at the same, homologous nucleotide positions. In the case of the 18S rRNA, 30 aligned sequences included 23 species and stretched over 1879 alignment positions, whereby different haplotypes of *S. arctica*, *S. caninum*, and *S. canis* were included to allow for variability and determine their potential influence on tree topology. The only publicly available 18S rRNA sequences of *S. felis* (AY190080, AY576489) were excluded because they were too short (361–696 bp) for the type of analysis performed in this study. For the 28S rRNA, a total of 24 sequences of 17 species were aligned, and the final dataset consisted of 1594 alignment positions. We aligned 31 ITS-1 sequences of 20 species over 2512 alignment positions. Owing to a relatively high degree of mismatch between the ITS-1 sequences (e.g., between *S. neurona*, AF252407, and related taxa and the rest), we allowed for a relatively high percentage of gaps at any one position (14–20%), so that between 424 and 625 homologous sites could be analyzed. The mitochondrial *cox*1 sequences were aligned via the program MUSCLE, which was embedded in MEGA 11 software. Twenty-seven aligned *cox*1 sequences of 22 *Sarcocystis* spp. were truncated at both ends, and 1053 aligned nucleotide positions were subjected to the phylogenetic analysis. Except for the ITS-1 sequences, all positions with less than 95% site coverage were eliminated, i.e., fewer than 5% alignment gaps, missing data, and ambiguous bases were allowed at any position (partial deletion option). The reliability of the resulting tree topology was tested by bootstrapping via 1000 replications.

For inference of phylogenetic trees, the Maximum Likelihood (ML) algorithm was used for all four marker genes, whereby the Tamura 3-parameter (18S rRNA, ITS-1) and Hasegawa–Kishino–Yano (28S rRNA, *cox*1) substitution models were employed. They were selected via the ML analysis as implemented in MEGA11 based on the lowest BIC (Bayesian information criterion) and AICc (Akaike information criterion, corrected) values. All the substitution models used a discrete Gamma distribution to model evolutionary rate differences among sites, whereas the rate variation model allowed for some sites to be evolutionarily invariable (+G+I), except in the case of *cox*1, where only a Gamma distribution was applied (+G). The GenBank accession numbers of all analyzed taxa are shown in the respective phylogenetic trees.

## 3. Results

### 3.1. Morphological Characteristics of Sarcocysts in Cheetahs

We observed sarcocysts in two of the six examined cheetahs (#5 and #6). In both animals, sarcocysts were present in the skeletal muscles, tongue, diaphragm, and esophagus, but not in the heart. Under the LM, there was apparently only one morphologic type of sarcocyst present in the two animals (Figure 1). The sarcocysts were slender (Figure 1a), septate (Figure 1b), and surrounded by a thin striated cyst wall that was 1.4–2.1 μm thick (Figure 1b,c). Mature sarcocysts measured 549–2720 × 119–178 μm (n = 20) in size and contained lancet-shaped cystozoites, which were 7.1–11.0 × 1.7–3.6 μm (n= 30) in size, and on average 9.1 ± 1.2 (SD) μm long and 2.7 ± 0.5 μm in diameter.

Four sarcocysts from both infected cheetahs were studied using TEM (Figure 1e,f), all of which had walls that were ultrastructurally similar and closely resembled type 9c of the commonly used cyst wall classification scheme [1]. Their wall consisted of a wavy parasitophorous vacuolar membrane, which had pleomorphic small (up to 1.7 × 0.5 μm) villous protrusions (Vp) that were lined with a smooth electron-dense layer (Edl). The Vp contained no microtubles but rather fine granular, electron-light material. A ground substance layer (Gsl) measuring 0.8–1.1 μm in thickness was located immediately beneath the sarcocyst wall. Numerous invaginations presented on the surface of Gsl. There were three rhoptries occupying the anterior of the cystozoities. When we compared the sarcocyst morphology of our samples from cheetahs with those from domestic and wild terrestrial carnivores published previously, particularly considering the ultrastructure of the sarcocyst wall, we could not identify any significant differences in *S. felis*, *S. caninum*, or *S. arctica* (Table 1).

### 3.2. Intraspecific and Interspecific Sequence Similarities of the Sarcocystis sp. from Cheetahs

The PCR amplification and sequencing of the four genetic markers were successful. The four nuclear 18S rRNA sequences were 1799–1804 bp long and shared 99.8–100% identity (average: 99.9%). The four 28S rRNA sequences were 3285–3287 bp long and were 99.5–99.7% identical (average: 99.6%). The four ITS-1 sequences were 947–949 bp in length and shared 99.1–99.8% identity (average: 99.3%). Finally, four mitochondrial *cox*1 sequences were 1085 bp long and shared 99.9–100% identity (average: 99.9%). All sequences were deposited in GenBank under the accession numbers OQ689797–OQ689799 (18S rRNA); OR436907–OR436910 (28S rRNA); OQ689800, OQ689801, OQ676521, and OQ676522 (ITS-1); and OQ726125 and PQ217791 (*cox*1).

When we compared the newly obtained sequences with those from closely related *Sarcocystis* species previously deposited in GenBank (Table 2), we observed the highest similarity with *S. caninum* and *S. arctica*, i.e., 99.9–100% and 99.8–100% identity for the 18S rRNA, 99.5% and 99.3–99.5% identity in the case of the 28S rRNA, 95.9–97.5% and 96.3–97.3% identity at the ITS-1 locus, and 99.6% and 99.2–99.7% identity regarding mitochondrial *cox*1, respectively. Thus, in three of the four marker genes, the identity values were ≥99% for these two species.

Variability was greater regarding the ITS-1 sequences, which also allowed for a direct comparison with various haplotypes of *S. felis*, for which no other genetic markers of appropriate length were available. The results of pairwise BLAST comparisons indicated that the *Sarcocystis* species under investigation was more distant from *S. felis* than *S. caninum* and *S. arctica*, with identity values ranging between 87.5 and 88.9% and 95.9 and 97.5%, respectively. Additionally, at this locus, the newly obtained sequences presented markedly lower identity values with those of *S. canis* and *S. svanai* (78.4–89.8% identity) than in the BLAST results of the 18S rRNA, where it was >99% identical.

### 3.3. Phylogenetic Trees

The results of our phylogenetic analyses are shown for ITS-1 (Figure 2a), mitochondrial *cox*1 (Figure 2b), 18S rRNA (Appendix A), and 28S rRNA (Appendix A) sequences. Regarding all four marker genes, the *Sarcocystis* species under investigation formed a well-supported clade including various haplotypes of *S. arctica* and *S. caninum* from various locations in North America and Europe, also including different species of carnivores as intermediate hosts such as *Canis lupus*, *C. familiaris*, *Vulpes lagopus*, and *V. vulpes*. This finding indicated that the parasites of this clade exhibited a relatively wide geographical distribution range involving different intermediate host species. In the 18S rRNA and *cox*1 trees, we additionally included samples of *S. arctica* from a putative definitive host (white-tailed eagle, *Haliaeetus albicilla*) to highlight the fact that these sequences (MZ329343 and MZ332957, respectively) were fully identical or almost identical to the other *S. arctica* isolates.

While the currently available 18S rRNA sequences of *S. felis* were too short for the phylogenetic analysis, our reconstruction of the ITS-1 phylogenetic tree revealed that this taxon formed a clade that assumed a sister position relative to *S. arctica*/*S. caninum* isolates (Figure 2a). Hence, at least with respect to ITS-1, *S. felis* is genetically different from *S. arctica* and *S. caninum*. Furthermore, all three genetic markers (ITS-1, 18S rRNA, *cox*1) for which sequences of *S. canis* were available revealed that *S. canis* assumed a position as a sister species relative to *S. felis* and *S. arctica*/*S. caninum*, branching off basally from the joint clade. Although branch support for the split of *S. canis* from the rest of the taxa was only moderate in the 18S rRNA tree, branch credibility for this topology was markedly higher in the ITS-1 (66–74%) and *cox*1 (88–90%) trees. *Sarcocystis svanai* (KM362428) assumed a somewhat ancestral position to the *S. arctica*/*S. caninum* isolates in the 18S rRNA tree, and this species clustered with *S. lutrae* in the ITS-1 phylogeny.

Overall, the phylogenetic results for all four genetic markers indicated that the *Sarcocystis* spp. infecting carnivores as intermediate hosts contributed one half to a monophyletic subclade that was characterized by species using birds as definitive hosts (e.g., birds of prey); the other half of that clade contained species with a bird-bird life cycle (e.g., *S. calchasi*, *S. halieti*, *S. lari*, *S. turdusi*, etc.). This dichotomy within the clade was especially apparent in the *cox*1 tree, where *S. lari* (using birds as definitive and intermediate hosts) is branching off at the base of the bird–bird species and *S. lutrae* (with a bird–mustelid life cycle) branches off at the base of the bird–carnivore *Sarcocystis* spp. In the ITS-1 tree, *S. lutrae* branched basally from the bird host clade. All four phylogenies also revealed that *S. neurona* and its closest relatives (i.e., *S. falcatula* and *S. speeri*) form a separate lineage, as do species that directly cluster with *S. rileyi*. Based on the genetic evidence from our results on pairwise sequence similarity (Table 2) and the phylogenetic reconstructions (Figure 2, Appendix A), we are confident that the *Sarcocystis* samples from the two cheetahs in China contained *S. arctica*.

## 4. Discussion

Based on the morphological and molecular results of our study, we propose that the sarcocysts isolated from the two cheetahs in China belong to *S. arctica*. This is the first record of this species in a felid host and adds a new location record beyond its currently known distribution in North America and Europe. Although the two infected cheetahs were imported from Africa, where they could have potentially acquired the parasite, it is equally likely that they became infected in China since the cheetahs lived in outdoor enclosures for sightseeing in zoological gardens. This new location and our molecular comparisons imply that *S. arctica* has a wide distribution range and uses different carnivore species as intermediate hosts, e.g., as evident from closely related haplotype sequences from different sampling locations. A large distribution range of the parasite could be indicative of a definitive host with a large distribution range. It is therefore not surprising that white-tailed sea eagles (*Haliaeetus albicilla*) [17], hooded crows (*Corvus cornix*), and common ravens (*Corvus corax*) [16] have been identified molecularly as potential definitive hosts of *S. arctica*.

We are aware that our interpretation of the data diverted from the traditional taxonomic approach to *Sarcocystis* spp. delimitation in carnivores, which mainly relies on sarcocyst morphology and host specificity. Accordingly, the sarcocysts previously found in feline carnivores were all attributed to one species, *S. felis*, including those found in cheetahs [4]. However, morphologically very similar sarcocysts that were found in canids were named as two separate *Sarcocystis* species, one, *S. arctica*, in arctic foxes [6], and the other, *S. caninum*, in domestic dogs [8]. While to the untrained eye ultrastructural differences may exist between the sarcocyst walls of these species, this should not distract from the fact that the sarcocysts of this lineage of *Sarcocystis* spp. are morphologically highly similar but pleomorphic in terms of their villous protrusions, sometimes showing different forms within a single sarcocyst, e.g., [7,8,29]. Despite the morphological similarities between all three species (i.e., *S. felis* in domestic cats [30,31], bobcats [4], Florida panthers and cougars [32], lions [5,33], and jaguarundi and southern tiger cats [14]; *S. arctica* in arctic foxes [6,7], red foxes [34,35], and Alaskan wolves [29]; and *S. caninum* in domestic dogs [8,10]) and the genetic identity between *S. caninum* and *S. arctica* as recognized for the ITS-1 locus [9,10], all three taxa have been regarded as valid species until today [1,11]. In view of our phylogenetic analysis of the ITS-1 locus, we agree that treating *S. felis* as a separate species may be justified because some of the available ITS-1 sequences are clearly distinct from those of *S. caninum* and *S. arctica*. On the other hand, our samples from cheetahs in China are not *S. felis* because the ITS-1 sequences from China show relatively high identity values with those from *S. arctica*, despite the fast evolution of ITS-1 (see scale bar in Figure 2a). In our study, *S. caninum* was molecularly identical to *S. arctica* with respect to all four genetic markers used here. Hence, we see no reason to further treat the two taxa as different species. Additionally, no significant morphological differences from *S. caninum* were discussed in the first ultrastructural description of the sarcocysts of *S. arctica*, which could justify a distinction [29]. Since *S. arctica* was described earlier than *S. caninum*, *S. caninum* could be treated as a junior synonym. While we acknowledge that cross-transmission experiments could certainly help clarify species delimitation [8], we are reluctant to follow the argument that morphologically and genetically similar *Sarcocystis* spp. could be distinguished only by this method. We believe that the repertoire of genetic markers for *Sarcocystis* species has broadened to a level where subtle differences between taxa can also be detected. Here, the use of ITS-1 for a group of closely related species that use mainly birds and carnivores as hosts may be one example; the use of the *cox*1 gene for the delimitation of *Sarcocystis* spp. in ruminants may be another [12]. The often-cited example of morphological and biological similarity between *S. tenella* and *S*. *capracanis*, which both share canids as definitive hosts but use different ruminant intermediate hosts and show no clear ultrastructural differences regarding their sarcocyst wall [1], also illustrates that the genetic markers such as 18S rRNA [36] and *cox*1 [37] could distinguish well between the two species, corroborating their biological differences in terms of intermediate host preference. In contrast, the finding of highly identical sequences in several species of carnivorous intermediate hosts, such as those in this study, suggests that only one species, *S. arctica*, is involved.

Among our samples from the cheetahs, only one morphological type of sarcocyst was observed. While the similarities regarding sarcocyst wall morphology have been outlined above, one must also note that the size of the cystozoites was almost identical in all three species (*S. arctica*, S*. caninum*, and *S. felis*), including the samples from cheetahs (with an average cystozoite length matching that of *S. arctica*). We believe that the diagnostic value of cystozoite size is underappreciated but can be useful, especially in the case of *Sarcocystis* spp. with a wide distribution range [38], where one would expect this trait to be rather constant. Furthermore, because we found that *S. canis*, for which the sarcocyst is still unknown [1,11], consistently clustered with the three species above in our molecular analyses, we hypothesize that its sarcocysts might be morphologically similar and that its definitive host may be a predatory or scavenging bird, or both. Previously, the positions of *S*. *canis* and *S*. *felis* in the phylogenetic tree were unclear, partly because of a lack of appropriate nucleotide sequences but also because of problems associated with small genetic distances between taxa, as mentioned in the Introduction. We are confident that our secondary structure-guided alignment of the ribosomal sequences produced the correct tree topologies, including ITS-1. Overall, the phylogenetic results suggest that the group of *Sarcocystis* spp. that use carnivores as intermediate hosts shares the most recent common ancestor with a cluster of species with a bird-bird life cycle. However, *Sarcocystis* spp. with a carnivore–bird life cycle, such as *S. rileyi* [39], apparently represent a separate lineage, as do the species clustering with *S. neurona*, which are highly divergent in our ITS-1 phylogenetic tree.

Finally, we would like to provide some hypothetical answers to the following question: how can *Sarcocystis* be transmitted between a predatory definitive and a predatory intermediate host? The occurrence of sarcocysts in the musculature of carnivorous mammals has been observed infrequently, with *Sarcocystis* spp. in focus here serving as prime examples. However, these are not the only examples where predatory animals themselves become intermediate hosts for *Sarcocystis*: muscular sarcocystosis also occurs in birds of prey (e.g., muscle cysts of *S. halieti* in birds of prey [40]) and in snakes (*Sarcocystis* in muscles of a South American rattlesnake [41]). Although one could explain these infections as an aberrant or unusual developmental path of the parasites, they make perfect sense if one strictly applies the predator-prey logic of the *Sarcocystis* life cycle because larger (apex) predators prey on smaller predators, and both types of predators may be eaten by scavengers after their death. For example, large eagles prey on smaller predatory birds and mammalian carnivores [42,43,44]. Intriguingly, a recent study spanning 23 years of countrywide data from nesting records of golden eagles (*Aquila chrysaetos*) and wildlife track counts of mesopredators in Finland revealed that foxes and martens were, against expectations, more abundantly close to eagle nesting sites [45]. In view of *Sarcocystis* infection, these data could explain how predatory felids, canids, or mustelids could acquire infection in nature via environmental contamination, with *Sarcocystis* sporocysts leaking from bird feces around nesting sites. The putative *Sarcocystis* spp. with a life cycle that could fit this environmental scenario are *S. arctica* and *S. lutrae*. The latter species uses a variety of small carnivores as intermediate hosts, including mustelids, and develops in the white-tailed eagle as a definitive host [46]. If scavenging birds such as *Corvus* spp. were added to this scenario, the number of potential parasite transmission routes could increase further. For example, wolves (*Canis lupus*) interact most with the common raven (*Corvus corax*) as a non-prey species in North America, from which the ravens benefit by scavenging wolf kills [47]. While potentially contaminating wolf kills with *Sarcocystis* sporocysts, out of this competition for food, there may arise occasions where wolves that died would be scavenged by ravens, closing the parasite’s life cycle if the wolf had carried sarcocysts in their musculature.

## 5. Conclusions

In this study, we considered the sarcocysts isolated from the musculature of two cheetahs in zoos in China to be *S. arctica*, which is the first time that a *Sarcocystis* species assumed to be specific for canids appeared in a felid intermediate host. Our morphological comparisons and phylogenetic analyses suggested that *S. arctica* and *S. caninum* are synonymous and *S. felis* is a separate species owing to its divergence from *S. caninum* and *S. arctica* at the ITS-1 locus. However, other genetic markers for *S. felis* need to be analyzed in the future to conclude its status. Based on our hypotheses on the ecological relationships among the hosts of the *Sarcocystis* species in focus, it might be worthwhile to investigate these through field studies, potentially uncovering the epidemiological factors that could lead to muscular sarcocystosis in carnivores. This may benefit contemporary classification, conservation, and disease control efforts.

## Figures and Tables

**Figure 1 animals-15-00180-f001:**
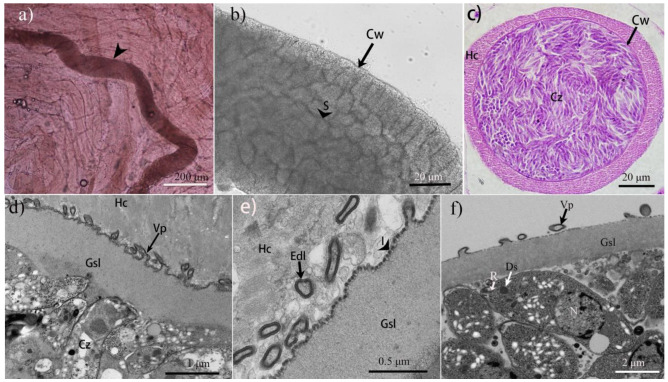
Morphological characteristics of the sarcocysts from the two cheetahs, which we address as *Sarcocystis arctica*, extracted from their skeletal muscles and examined via light microscopy (LM, **a**–**c**) and transmission electron microscopy (TEM, **d**–**f**). (**a**) A sarcocyst (arrowhead) in skeletal muscle (unstained). (**b**) A sarcocyst excised and freed from muscle tissue (unstained). Note the thin striated cyst wall (Cw) on the surface of the sarcocyst, and the septate (S, arrowhead) interior of the sarcocyst. (**c**) The cross-section of a sarcocyst (stained with hematoxylin and eosin). The sarcocyst is surrounded by a host cell (Hc), and filled with cystozoites (Cz). Note the thin striated cyst wall (Cw). (**d**) The longitudinal section of a sarcocyst. Pleomorphic small villous protrusions (Vp) on the sarcocyst wall. Note the ground substance layer (Gsl) under the surface of the cyst wall and Cz within the sarcocysts. (**e**) The diagonal section of a sarcocyst. A smooth electron-dense layer (Edl, arrow) lining the Vp, and invaginations (I, arrowhead) delimiting the Gsl on the surface in the space between Vp. (**f**) The diagonal section of a sarcocyst. Note the rhoptries (Rs), electron-dense granules (Ds), and a nucleolus within Cz.

**Figure 2 animals-15-00180-f002:**
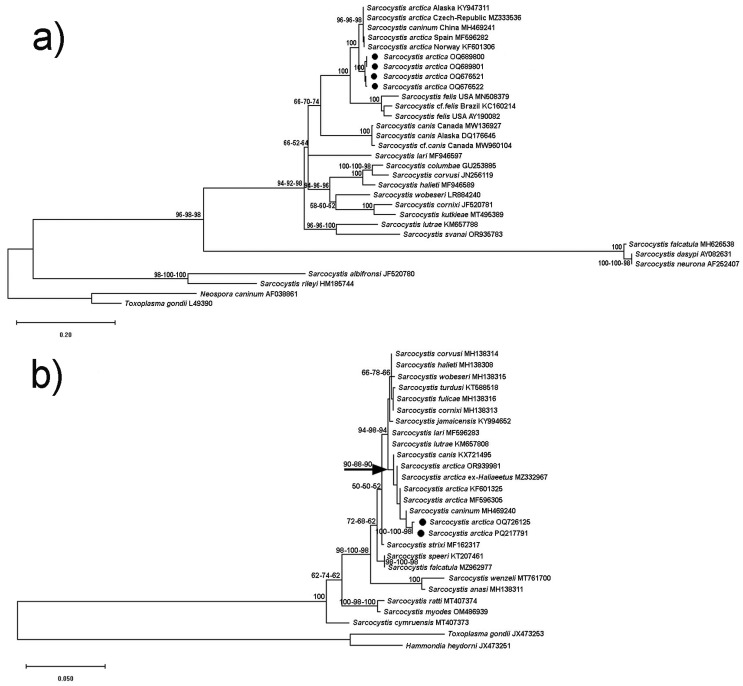
Maximum Likelihood (ML) phylogenetic trees inferred from nuclear ITS-1 (**a**) and mitochondrial *cox*1 (**b**) sequences. The GenBank accession numbers of all the sequences included in the analyses are given beside the taxon names. The values attached to branches represent percent bootstrap values per 1000 replicates (only values ≥50% are shown), whereby the results of three independent analyses (i.e., alignments) are indicated. Species of Toxoplasmatinae served as outgroups. The new nucleotide sequences of *Sarcocystis arctica* from cheetahs are highlighted by black symbols. Species with birds or carnivorous mammals as definitive hosts, including species closely related to *S. neurona*, were included. However, *Sarcocystis* spp. of ruminants as intermediate hosts were excluded for reasons of topological stability of the phylogenetic trees (see methods).

**Table 1 animals-15-00180-t001:** Important morphological traits of the sarcocysts observed in cheetahs in comparison with those of *S. arctica*, *S. caninum*, and *S. felis* based on ultrastructural data (cyst wall) and light microscopical measurements (cystozoites and sarcocyst size).

Morphological Trait	*Sarcocystis* ex, Cheetah, China *	*S. arctica* *	*S. caninum*	*S. felis*
Sarcocyst wall type	9c	9c	9c	9c
Appearance of Vp	Pleomorphic Vp, wavy wall, no microtubules	Pleomorphic Vp, no microtubules, some anastomosing	Elongated Vp, wavy wall, no microtubules	Hobnail-like bumps and Vp, no microtubules
Size of Vp in μm	1.7 × 0.5	1.5 × 0.5	1.0 × 0.8	0.6–1.2 × 0.3–0.4
Sarcocyst size	0.5–2.7 mm long, 120–180 μm wide	Up to 12 mm long, 250 μm wide	Up to 1.2 mm long, 75 μm wide	2.1 mm long, 150 μm wide
Sarcocyst wall, thickness in μm (Gsl + Vp)	1.4–2.1	Up to 3.5	Up to 2.0	1.0–1.5
Cystozoites *, size in μm (length × width)	7.1–11.0 [av. 9.1] × 1.7–3.6 [av. 2.7]	Ex *Canis lupus*, 9.4–9.8 [av. 9.5] × 1.3–1.6 [av. 1.5]; ex *Vulpes lagopus*, 6.9–9.5 × 1.4–2.7	7.5–9.0 long	7.0–10.0 × 1.5

Vp: villous protrusions; Gsl: ground substance layer. * Except for the samples of *Sarcocystis* from cheetahs and *S. arctic*a from *Vulpes lagopus* where measurements relate to live cystozoites, measurements of the other species used fixed cystozoites processed for microscopy (LM, TEM); data on *S. arctica* are from Cerqueira-Cézar et al. (2017) [7] and Calero-Bernal et al. (2016) [29], *S. caninum* from Dubey et al. (2015) [8], and *S. felis* from Dubey et al. (1992) [4].

**Table 2 animals-15-00180-t002:** Interspecific nucleotide sequence comparison similarities between the *Sarcocystis* sp. from cheetahs and closely related taxa previously deposited in GenBank.

DNA Regions	Accession Numbers (Length, bp)	Identity with Sequences Previously Deposited in GenBank		
		*Sarcocystis* sp. (Accession Numbers)	% Query Coverage	% Identity (Average)
18S rRNA	OQ689797–OQ689799 (1803–1804)	*S. caninum* (MH469238, MH469239, KM362427)	92–100	99.9–100 (99.8)
		*S. arctica* (MZ329343, KX022100–KX022102, MF59623717–MF596237, KF601301)	92–100	99.8–100 (99.9)
		*S. svanai* (KM362428)	92	99.4–99.5 (99.5)
		*S. columbae* (GU253883)	90	99.4
		*S. felis* (AY190080, AY576489)	20–39	99.2–99.4 (99.3)
		*S. canis* from *Ursus americanus* and *Zalophus californianus* (OR654898, OR339987)	88	99.1
28S rRNA	OR436907–OR436910 (3285–3287)	*S. caninum* (MH469239)	100	99.5
		*S. arctica* (KY947309, MF596240–MF596260, KX22104–KX22107, KY609323, KY947308)	44–45	99.3–99.5 (99.4)
		*S.* (*Frenkelia*) *glareoli* (AF044251)	100	98.3
ITS-1	OQ689800, OQ689801, OQ676521, OQ676522 (948–949)	*S. caninum* (MH469241, JX993923)	100	95.9–97.5 (96.7)
		*S. arctica* (MZ333536, KY947311, KF601306, KX022108–KX022111, KF601308, KY947310, OK481372–OK481376, KX156837, MF596262–MF596282)	49–92	96.3–97.3 (96.3)
		*S. felis* from domestic cats (AY190081, AY190082, MN508375–MN508379)	35–79	87.8–88.9 (88.3)
		*S. felis* from wild felids (KC160213, KC160214)	48–72	87.5–88.5 (88.0)
		*S. canis* from *Ursus* spp. (DQ176645, MW960104)	83–86	89.6–89.8 (89.7)
		*S. svanai* (OR935783)	58	78.4
		*S. lari* (MF946597, MN450357)	49–97	76.1–88.3
		*S. lutrae* (KM657788, KM657805)	86	84.9
*cox*1	OQ726125, PQ217791 (1085)	*S. caninum* (MH469240)	100	99.6
		*S. arctica* (KX022112–KX022115, KY947304, KY947305, KY609324, MF596286–MF596306, KF601318–KF601321, MZ332967)	85–96	99.2–99.7 (99.5)
		*S. lari* (MF596283, MF946584)	96	99.1
		*S. lutrae* (KM657808)	97	98.8

## Data Availability

All relevant data of this study are contained within the manuscript, and the datasets used and/or analyzed during the current study are available from the corresponding author (JH) upon reasonable request. Nucleotide sequences of the 18S rRNA (OQ689797–OQ689799), 28S rRNA (OR436907–OR436910), ITS-1 (OQ689800, OQ689801, OQ676521, and OQ676522), and mitochondrial *cox*1 (OQ726125 and PQ217791) of the samples have been deposited in GenBank.

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
