# Peer review of "Morphological and Molecular Identification of Sarcocystis arctica in Captive Cheetahs (Acinonyx jubatus) in China Helps Clarify Phylogenetic Relationships with Sarcocystis caninum and Sarcocystis felis"

_animals, 2025, doi:10.3390/ani15020180_

Round 1
Reviewer 1 Report
Comments and Suggestions for Authors
General comments: The manuscript is well written and structured, with interesting and original results. The English and grammar need to be revised, especially in the introduction. Furthermore, after minor changes, I recommend publication.
Introduction:
Line 53: This definition of the Sarcocystis cycle based on the prey (IH):predator (DH) ratio, despite being in the literature, is questionable and should be avoided. The article itself contains examples of felines (predators) that are IH of this agent (Line 64), not fitting this definition.
Line 87: I suggest changing the term "captive" to "under human care", which is a more appropriate term. I recommend replacing it throughout the manuscript.
Materials and Methods:
Line 104: If the animals were necropsied, it would be interesting to include the results of the necropsy, and the indication of whether or not there was a relationship with parasitism by Sarcocystis.
Line 206: Correct the formatting of scientific names, which should be in italics whenever they are cited. This correction must be made throughout the manuscript, as at several points the scientific name does not appear in italics.
Figure 2: The image of the phylogenetic tree is very small and has low resolution. In order to see the data clearly, it is necessary to enlarge it on the computer (which is not desirable). I suggest choosing to display only the most important trees (ITS and cox), leaving the others as complementary figures. Another question in this figure is why the four amplified samples from the study are not listed in the Cox and 18S tree.
Author Response
General comments: The manuscript is well written and structured, with interesting and original results. The English and grammar need to be revised, especially in the introduction. Furthermore, after minor changes, I recommend publication.
Introduction:
Line 53: This definition of the Sarcocystis cycle based on the prey (IH):predator (DH) ratio, despite being in the literature, is questionable and should be avoided. The article itself contains examples of felines (predators) that are IH of this agent (Line 64), not fitting this definition.
Response: According to the suggestion. The sentence has been edited as “Species of the genus Sarcocystis are tissue cyst-forming intracellular protozoan parasites with an obligate, heterogeneous life cycle that is based on a predator-prey relationship between a definitive host (predator) and an intermediate host (prey)” in lines 51-53 in the newly edited ms . We need to explain that it is a predator (DH) - prey (IH) relationship, not a IH/DH relationship. The definitive host comes FIRST because it includes the sexual development which was there earlier in evolutionary times (monoxenous coccidian ancestors).
Line 87: I suggest changing the term "captive" to "under human care", which is a more appropriate term. I recommend replacing it throughout the manuscript.
Response: The captive has been frequently used in other references. We think it is suitable. Therefore the suggestion has not been adopted. Thank you .
Materials and Methods:
Line 104: If the animals were necropsied, it would be interesting to include the results of the necropsy, and the indication of whether or not there was a relationship with parasitism by Sarcocystis.
Response: The results of pathological examination has been published by orhers, and here we added the result in Lines 104-112 of materials: The pathological findings have been published by others: Briefly, histopathological examination revealed chronic renal insufficiency (4/5), interstitial pneumonia (2/5), necrotic splenitis (1/5), splenatrophy (2/5), septic spleen (1/5), systemic atrophy (1/5), arterial sclerosis (1/5), and acute endometritis (female cheetah), whereby three of the five males were diagnosed to have died of renal insufficiency and the other two by natural aging. Additionally, serological examination indicated that all six animals were infected with Toxoplasma gondii and viable Toxoplasma parasites could be retrieved from muscle tissue through bioassays in mice. However, none of the cheetahs showed any acute parasite infection [18].
Line 206: Correct the formatting of scientific names, which should be in italics whenever they are cited. This correction must be made throughout the manuscript, as at several points the scientific name does not appear in italics.
Resonse: we agree, the writting mistake has been corrected throughout the ms.
Figure 2: The image of the phylogenetic tree is very small and has low resolution. In order to see the data clearly, it is necessary to enlarge it on the computer (which is not desirable). I suggest choosing to display only the most important trees (ITS and cox), leaving the others as complementary figures. Another question in this figure is why the four amplified samples from the study are not listed in the Cox and 18S tree.
Response: we accept this sugggesion. The resolution of fig 2 has been improved. According to the suggestion, the ITS-1 and cox1 trees have been reserved, and the other trees are served as supplementary materials. In the study, if the sequences of clones are completely identical, for avoiding redundant, we only submitted the different sequences to the GenBank. It is the reason that only three 18 rRNA sequences and two cox1 sequences were listed in the manuscript. Actually, in the Line 250 and 253, while we compared the newly obtained sequences, the identity of 18S rRNA (99.8–100%) and cox1 (99.9–100%) sequences is a hint that not all sequences were needed to be provided to GenBank.
Reviewer 2 Report
Comments and Suggestions for Authors
1) In conservaiion biology, cheetahs are highly endangerous species, so as eaven captive iindividuals, the authors sshould add their causues of the animalsi in brief to the part "Materials". If a report of the postmortem examinations has been published by veterinary officers in those zoos, they show them there.
2) They conclued "specific for canids as intermediate hosts to infect a feline host", so the authors show the arrangement of captiive canidss in each zoo.
3) The present MS was regarded not only as protozoological resserch, but as veterinary medicine. Hence, follow the comments from the pressent rferee, and after then, it should be published ASAP.
Author Response
In conservaiion biology, cheetahs are highly endangerous species, so as eaven captive iindividuals, the authors sshould add their causues of the animalsi in brief to the part "Materials". If a report of the postmortem examinations has been published by veterinary officers in those zoos, they show them there.
Response: The postmortem examinations has been added in lines 104-112 of the newly edited manuscript. That is “The pathological findings have been published by others: Briefly, histopathological examination revealed chronic renal insufficiency (4/5), interstitial pneumonia (2/5), necrotic splenitis (1/5), splenatrophy (2/5), septic spleen (1/5), systemic atrophy (1/5), arterial sclerosis (1/5), and acute endometritis (female cheetah), whereby three of the five males were diagnosed to have died of renal insufficiency and the other two by natural aging. Additionally, serological examination indicated that all six animals were infected with Toxoplasma gondii and viable Toxoplasma parasites could be retrieved from muscle tissue through bioassays in mice. However, none of the cheetahs showed any acute parasite infection [18]”.
They conclued "specific for canids as intermediate hosts to infect a feline host", so the authors show the arrangement of captiive canidss in each zoo.
Response: it is not possible that a canid infected with Sarcocystis in the musculature can infect a felid in the same zoo. The DH distributes the infective material...here, probably a bird.
The present MS was regarded not only as protozoological resserch, but as veterinary medicine. Hence, follow the comments from the pressent rferee, and after then, it should be published ASAP.
Response: I do not agree to make it a veterinary case. Again, where is the data to do that?
Reviewer 3 Report
Comments and Suggestions for Authors
The authors presented their work with the title "First identification of Sarcocystis arctica in captive cheetahs 2 (Acinonyx jubatus) in China with annotations on closely related 3 species in carnivorous intermediate hosts".
The topic is very interesting but my opinion is that the phylogenetic findings are more exciting than detection of S. arctica or S. caninum in cheetah. (Though even this finding is also interesting.)
I recommend rewording of the manuscript with different focus. In my viewpoint, the most important thought of the manuscript can be found in Line 369-370.
Minor comments:
Line 57-59 - redundant information
Line 78 - In reference [14], the topic of the sentence cannot be found.
Line 344 - In reference [30], the topic of the sentence cannot be found.
Based on these research finding, it is worth recommending rethinking of Sarcocytis spp. taxonomy.
Author Response
The authors presented their work with the title "First identification of Sarcocystis arctica in captive cheetahs 2 (Acinonyx jubatus) in China with annotations on closely related 3 species in carnivorous intermediate hosts".
The topic is very interesting but my opinion is that the phylogenetic findings are more exciting than detection of S. arctica or S. caninum in cheetah. (Though even this finding is also interesting.)
I recommend rewording of the manuscript with different focus. In my viewpoint, the most important thought of the manuscript can be found in Line 369-370.
Response: we fully agree with this reviewer for he fullly understands and appreciates the significance of our findings.
Minor comments:
Line 57-59 - redundant information
Resposne: the sentence is the background of cheetah. We think it is suitable to be reserved.
Line 78 - In reference [14], the topic of the sentence cannot be found.
Resposne: The reference relates to the definition of the S. neurona cluster of species, which is mentioned several times in reference 14 (reference 13 in the newly edited manuscript). For instance, that is on page 187 (first paragraph, where taxa of this clade are mentioned) and in the discussion on page 195 (first paragraph of right column), where the 'S. neurona clade of species' is mentioned. Furthermore, Fig.6 B of that publication shows for the cox1 locus that the S. neurona clade exhibits lower evolutionary rates regarding amino acid changes (or, smaller genetic distances) than other clades of Sarcocystis.
"Line 344 - In reference [30], the topic of the sentence cannot be found."
Response: That S. arctica was reported to develop in the white-tailed eagle is already mentioned in the introduction of reference 30. However, the reviewer is correct in that this is not the original publication on the topic. Therefore we added the original reference here: Máca O, González-Solís D. Role of three bird species in the life cycle of two Sarcocystis spp. (Apicomplexa, Sarcocystidae) in the Czech Republic. Int J Parasitol Parasites Wildl. (2022) 17:133–7. doi: 10.1016/j.ijppaw.2022.01.002
Based on these research finding, it is worth recommending rethinking of Sarcocytis spp. taxonomy.
Response: The thought has been added in the conclusion.